# MicroRNA-155 is essential for the optimal proliferation and survival of plasmablast B cells

Giuseppina Arbore[1],*, Tom Henley[1],*, Laura Biggins[3], Simon Andrews[3], Elena Vigorito[1], Martin Turner[1], Rebecca Leyland[1,2]

**A fast antibody response can be critical to contain rapidly dividing pathogens. This can be achieved by the expansion of antigen-specific B cells in response to T-cell help followed by differentiation into plasmablasts. MicroRNA-155 (miR-155) is required for optimal T-cell–dependent extrafollicular responses via regulation of PU.1, although the cellular processes underlying this defect are largely unknown. Here, we show that miR-155 regulates the early expansion of B-blasts and later on the survival and proliferation of plasmablasts in a B-cell–intrinsic manner, by tracking antigen-specific B cells in vivo since the onset of antigen stimulation. In agreement, comparative analysis of the transcriptome of miR-155–sufficient and miR-155–deficient plasmablasts at the peak of the response showed that the main processes regulated by miR-155 were DNA metabolic process, DNA replication, and cell cycle. Thus, miR-155 controls the extent of the extrafollicular response by regulating the survival and proliferation of B-blasts, plasmablasts and, consequently, antibody production.**

## Introduction

Optimal humoral responses against foreign T-dependent antigens require crosstalk between B cells and CD4[+] T cells. After the binding of B cells to their cognate antigen, B cells localise to the B:T border, where they receive T-cell help. This interaction promotes extensive cell division and the migration of B cells to the B-cell follicles. Later on, the highly proliferative B-cell blasts differentiate into germinal centre cells or antibody-secreting cells (plasmablasts). These rapidly emerging plasmablasts are found in the extrafollicular tissue where they continue to expand until they cease proliferation and enter apoptosis (Maclennan et al, 2003; Tellier & Nutt, 2019). The

ability of B cells to quickly differentiate into short-lived antibody-secreting cells to produce neutralising antibodies of different isotypes can be critical to contain the spread of infections (Luther et al, 1997). Among the genes that regulate the extrafollicular response in a B-cell–intrinsic manner is microRNA-155 (*Mir-155*) (Vigorito et al, 2007, 2013). We previously showed that mice lacking miR-155 in B cells produce a lower number of IgM- and IgG-secreting plasmablasts relative to their wild-type counterparts (Vigorito et al, 2007). Furthermore, we identified PU.1 as a key miR-155 target for this process (Lu et al, 2014). However, whether the loss of cellularity of miR-155–deficient plasmablasts is due to a differentiation block, impaired proliferation or survival remains to be understood. Here, we address this issue by tracking miR-155[−/−]–activated B cells in vivo at critical stages of the extrafollicular response. The earliest impairment of antigen stimulated miR-155–deficient B cells was observed at day 2.5, at the time that B cells receive T-cell help, which precedes plasmablast differentiation. From this time point onwards, miR-155–deficient B-blasts and later on plasmablasts displayed defective proliferation and increased apoptosis. Gene expression analysis of miR-155–deficient plasmablasts revealed dysregulation of genes involved in proliferation, including DNA replication, cell cycle progression, and chromatin organisation. Overall, our data demonstrate a complex mechanism of plasmablast regulation by a single microRNA, which provides new insight into antibody production during the early response to infection.

## Results and Discussion

We previously showed that miR-155 is critical to sustain an efficient extrafollicular response in a B-cell–intrinsic manner and that this can be attributed in part to miR-155 regulation of PU.1 (Lu et al, 2014). To further understand the cellular basis by which miR-155 regulates the plasmablast response, we used the SW_{HEL} adoptive

[1]Lymphocyte Signalling and Development, Babraham Institute, Cambridge, UK    [2]Biomolecular Sciences Research Centre, Sheffield Hallam University, Sheffield, UK    [3]Bioinformatics, Babraham Institute, Cambridge, UK

Correspondence: R.Leyland@shu.ac.uk
Giuseppina Arbore's present address is Division of Immunology, Transplantation and Infectious Diseases, San Raffaele Scientific Institute, Milan, Italy.
Tom Henley's present address is Empyrean Therapeutics Ltd, Building 250, Babraham Research Campus, Cambridge, UK.
Elena Vigorito's present address is MRC Biostatistics Unit, School of Clinical Medicine, Cambridge Institute of Public Health, Cambridge Biomedical Campus, Cambridge, UK.
*Giuseppina Arbore and Tom Henley contributed equally to this work.

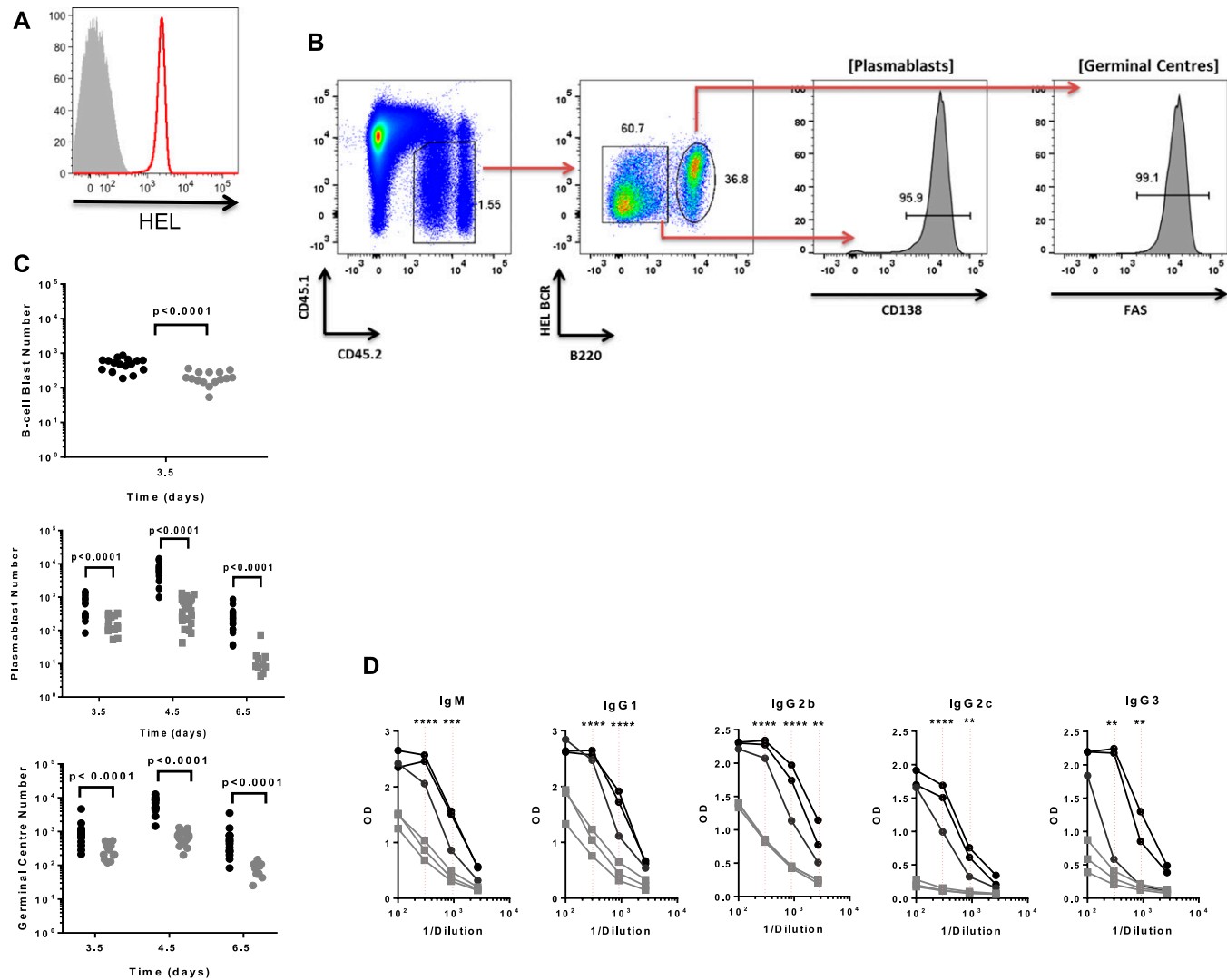

**Figure 1. miR-155 is required to sustain the plasmablast B-cell response.**
**(A)** A representative histogram showing HEL expression level on conjugated HEL-SRBCs (red) compared with unstained control (grey). **(B)** Representative flow cytometric plot showing gating strategy for SW$_{HEL}$ *Mir155*$^{+/+}$ B cells at days 4.5 post immunisation, for identification of CD45.2$^+$ donor derived HEL BCR$^+$, B220$^{lo}$ plasmablast B cells or HEL BCR$^+$, B220$^{hi}$ germinal centre B cells. **(C)** The number of SW$_{HEL}$ *Mir155*$^{+/+}$ (black) or *Mir155*$^{-/-}$ (grey) HEL-specific B-cell blasts, plasmablast B cells and germinal centre B cells was calculated per 10$^6$ lymphocytes after immunisation in mice (N = 16–19 independent *Mir155*$^{+/+}$ samples and 10–24 independent *Mir155*$^{-/-}$ samples). Data are representative of at least two independent experiments. For B-cell blast data, a Welch's *t* test was used. For plasmablast and germinal centre data, *P*-values are from *t* tests using the error mean square from the ANOVA. **(D)** HEL-specific antibodies of the indicated immunoglobulins were measured in the serum of mice injected with SW$_{HEL}$ *Mir155*$^{+/+}$ (black) or *Mir155*$^{-/-}$ (grey) B cells, at day 4.5 post immunisation with HEL-SRBCs. Red dotted line represents statistical analysis of indicated *Mir155*$^{+/+}$ or *Mir155*$^{-/-}$ values using two-way ANOVA with Sidak's multiple comparison test where **$P < 0.01$, ***$P < 0.001$, ****$P < 0.0001$.

transfer system (Phan et al, 2005). SW$_{HEL}$ transgenic B cells bear a rearranged hen egg lysosome (HEL)–specific VDJ$_H$ element targeted into the IgH chain locus combined with an HEL-specific k L-chain transgene (Phan et al, 2005). CD45.2$^+$ *Mir-155*$^{+/+}$ or *Mir155*$^{-/-}$ SW$_{HEL}$ B cells were adoptively transferred into wild-type CD45.1$^+$ congenic recipients and immunised with HEL coupled to sheep red blood cells (HEL-SRBCs—Fig 1A) to promote a T-dependent response.

We started by measuring the effect of miR-155 on the kinetics of the B-cell response. In the SW$_{HEL}$ system, B-cell blasts can be detected in the periarteriolar lymphoid sheath as early as 1 d after HEL-SRBC immunisation and commence proliferation from 1.5 d (Chan et al, 2009), and plasmablasts can be detected at day 3.5,

they peak by day 4.5 and rapidly decline afterwards (Paus et al, 2006; Phan et al, 2005). Adoptively transferred miR-155–sufficient or miR-155–deficient splenic B cells were stained for HEL B-cell receptor (BCR) in combination with CD45.1, CD45.2, CD138, FAS, and B220 and quantified using flow cytometry. In accordance with previous phenotypic characterisation of B-cell populations in the SW$_{HEL}$ system (Chan et al, 2009), B-cell blasts were detected as HEL binding, B220$^+$ cells during the early phase of the response and later plasmablasts were identified as HEL BCR$^+$, B220$^{lo}$. Plasmablast B cells have previously been shown to be Blimp-1$^+$ (Chan et al, 2009) and virtually all of these cells also expressed CD138 (Fig 1B). In addition, germinal centre B cells were detected as HEL BCR$^+$, B220$^{hi}$. These cells were

also shown to be virtually all FAS$^+$ (Fig 1B). In the absence of miR-155, we found a significant reduction in the number of B-cell blasts at day 3.5 post immunisation (Fig 1C). In addition, there was an almost 70% decrease in the mean number of $Mir155^{-/-}$ plasmablasts at day 3.5 compared with $Mir155^{+/+}$ cells, which became more severe at days 4.5 and 6.5 when the mean number of $Mir155^{-/-}$ plasmablasts decreased by more than 90% compared with $Mir155^{+/+}$ plasmablasts (Fig 1C). Although the peak of the response in $Mir155^{-/-}$ plasmablast B cells was also at day 4.5, it dramatically collapsed by day 6.5, a finding similar to that observed in the absence of IL-21 receptor in activated SW$_{HEL}$ B cells (Lee et al, 2011). We also observed a reduction in the number of germinal centre B cells in the absence of miR-155 at days 3.5, 4.5, and 6.5 (Fig 1C). In support of our observations, the production of HEL-specific antibodies of all subclasses examined, IgM, IgG1, IgG2b, IgG2c, and IgG3 isotypes, at day 4.5 was also significantly decreased in the absence of miR-155 (Fig 1D). At this time point, most of the antigen-specific antibody in serum is expected to be secreted by plasmablasts (Chan et al, 2009). These results indicate that the differentiation programme from activated B cells to extrafollicular plasmablasts is not abolished in the absence of miR-155; however, the number of differentiated cells is severely impaired resulting in a contracted and suboptimal response.

After establishing that miR-155 was critical for the plasmablast B-cell response, we next sought to determine the underlying cellular mechanisms. We started by monitoring the proliferation of antigen specific B cells. Division of antigen-specific B-cell blasts was assessed by CFSE dilution of HEL-binding, B220$^+$ cells every 24 h from 0.5 to 3.5 d post immunisation, a time when CFSE completely dilutes out. B cells lacking miR-155 showed a reduced proportion of highly dividing cells compared with wild-type at 2.5 d post immunisation, an effect which continued at day 3.5 (Fig 2A and B). The reduced number of proliferating cells could be due to a defect in cell cycle or apoptosis or both. To disentangle this, we analysed the cell cycle profile of SW$_{HEL}$ plasmablast B cells either sufficient or deficient in miR-155 at days 3.5 and 4.5 post immunisation. We chose those time points to analyse because of the onset of the plasmablast response at day 3.5 and because the number of $Mir155^{-/-}$ plasmablasts is highest at day 4.5, allowing robust detection. When DNA was quantified with DAPI, at day 3.5, there was a lower frequency of cells in the S-G2-M stage of the cell cycle in SW$_{HEL}$ $Mir-155^{-/-}$ plasmablast B cells compared with SW$_{HEL}$ $Mir-155^{+/+}$ plasmablasts and an increase in the frequency of cells in the G1 stage, which became statistically significant at day 4.5 (Fig 2C and D). Supporting this observation, we also measured the amount of plasmablast cells undergoing DNA replication at days 3.5, 4, and 4.5 by administering a pulse of the thymidine analogue 5-ethynyl-2′-deoxyuridine (EdU). SW$_{HEL}$ $Mir155^{-/-}$ plasmablast B cells at all time points exhibited decreased incorporation of EdU compared with SW$_{HEL}$ $Mir155^{+/+}$ plasmablasts, suggesting a defect in DNA replication (Fig 2E and F). This indicated a requirement for miR-155 in the progression between the G1 and S phases of the cell cycle.

Next, we asked whether miR-155–deficient B cells were undergoing increased apoptosis compared with their wild-type counterparts. Flow cytometry was used to analyse the frequency of active caspases in miR-155–sufficient or miR-155–deficient SW$_{HEL}$ plasmablast B cells at 3.5 and 4.5 d post HEL-SRBC immunisation.

We found that $Mir155^{-/-}$ plasmablasts expressed a significantly increased proportion of active caspases compared with $Mir155^{+/+}$ plasmablasts at both 3.5 and 4.5 d after activation (Fig 3A and B). The magnitude of the increase in caspase expression was greater as plasmablast differentiation progressed from day 3.5 to day 4.5. In an attempt to rescue the number of $Mir155^{-/-}$ plasmablasts by blocking apoptosis, we crossed the SW$_{HEL}$ $Mir155^{+/+}$ or SW$_{HEL}$ $Mir155^{-/-}$ mice with human $Bcl2$ transgenic mice, which express human BCL2 in all haematopoietic cells (Ogilvy et al, 1999). We found that the $Bcl2$ transgene was expressed equivalently in $Mir155^{+/+}$ or $Mir155^{-/-}$ plasmablasts (Fig 3C). After adoptive transfer and HEL-SRBC immunisation, the presence of the $Bcl2$ transgene substantially increased the relative frequency (Fig 3D) and number (Fig 3E) of miR-155–deficient plasmablasts. The mean number of SW$_{HEL}$ $Mir-155^{+/+}$ plasmablast B cells increased 1.3-fold in the presence of the $Bcl2$ transgene, whereas the number of SW$_{HEL}$ $Mir-155^{-/-}$ plasmablast B cells increased 2.5-fold in the presence of the $Bcl2$ transgene, further supporting a role for miR-155 in maintenance of plasmablast survival. However, despite a partial rescue in the number of miR-155–deficient plasmablasts, the number was not fully restored to wild-type levels, even when the frequency of active caspases was reduced (Fig 3F). These data suggest that the phenotype observed in miR-155–deficient plasmablast B cells is due in part to apoptosis of these cells, but that other mechanisms also play a role.

We next sought to determine the molecular pathways disrupted in SW$_{HEL}$ $Mir155^{-/-}$ plasmablast B cells by comparing the transcriptome to wild-type counterparts. CD45.2$^+$ HEL BCR$^+$ B220$^{lo}$ plasmablast B cells either sufficient or deficient for miR-155 were sorted to more than 98% purity (Fig 4A) and their transcriptome analysed by microarray at 4.5 d post immunisation. We defined differentially expressed genes as those genes with a fold change of at least 1.3 between $Mir155^{-/-}$ and $Mir155^{+/+}$ plasmablasts and a corrected $P$-value of less than 0.05. We observed 410 genes with increased mRNA abundance and 652 genes with decreased mRNA abundance in $Mir155^{-/-}$ plasmablasts relative to their wild-type counterparts. We then use the gene ontology (GO) enrichment analysis tool GOrilla (Eden et al, 2009) to look for pathway enrichment in the differentially expressed genes. Decreased and increased genes were sorted into functional processes and ranked according to their $P$-value (Tables S1 and S2). One of the most significantly decreased processes was translation. In addition, DNA replication and cell cycle were amongst some of the other most significant gene sets which were decreased in $Mir155^{-/-}$ plasmablasts (Fig 4B), data which also complimented our flow cytometric results. We observed overall the highest number of down-regulated genes in total associated with cellular and metabolic processes; however, the highest percentage of differentially regulated genes was shown to be largely associated with DNA replication and cell cycle mechanisms. For example, DNA unwinding, DNA strand elongation, and nucleosome organisation showed that 62.5%, 71.43%, and 77.27% of genes were differentially expressed, respectively (Table S1). We validated the diminished expression of selected genes involved in DNA replication and cell cycle, including c-myc, E2F1, E2F2, and c-myb, by RT-qPCR and found that in $Mir155^{-/-}$ plasmablasts, the expression of each was significantly lower than in wild-type cells (Fig 4C). Genes with increased mRNA abundance were

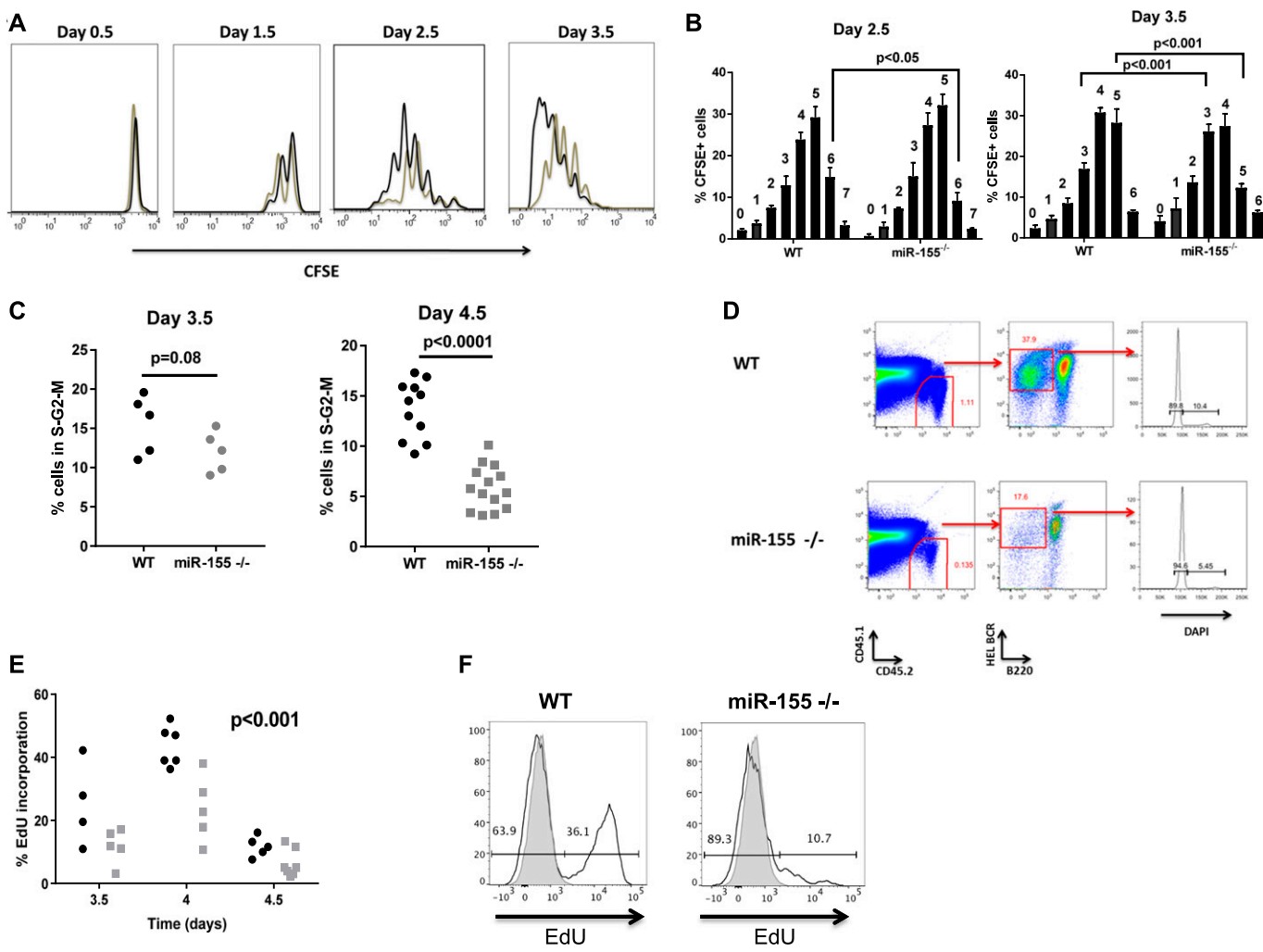

**Figure 2. miR-155 is essential for the optimal proliferation of plasmablast B cells.**
**(A)** $SW_{HEL}$ $Mir155^{+/+}$ (black line) or $Mir155^{-/-}$ (grey line) HEL-binding, B220$^+$ B cell blasts were assessed for different CFSE dilutions. **(B)** The percentage of CFSE$^+$ cells in each division at days 2.5 and 3.5 post immunisation. The generation number is indicated above each data set. A two-way ANOVA with Sidak's multiple comparison test was used. Values shown are mean ± SD. **(C)** Cell cycle analysis using DAPI staining was carried out on $SW_{HEL}$ $Mir155^{+/+}$ or $Mir155^{-/-}$ plasmablast B cells at 3.5 and 4.5 d post immunisation. The frequency of cells in the S-G2-M phases in $Mir155^{+/+}$ (black) or $Mir155^{-/-}$ (grey) plasmablasts is shown. A nonparametric Mann–Whitney test was used per time point. **(D)** Representative flow cytometry plot of DAPI staining at 4.5 d post immunisation in miR-155–sufficient or miR-155–deficient plasmablast B cells. Cells previously gated on lymphocytes and single cells. **(E)** EdU incorporation in $SW_{HEL}$ $Mir155^{+/+}$ (black) or $Mir155^{-/-}$ (grey) plasmablast B cells at the time points indicated post immunisation. Two-way ANOVA was used. There was no significant interaction between genotype and day, but both genotype and day had significant effects $P < 0.001$. **(F)** Representative flow cytometry histograms of EdU incorporation in wild-type (WT, black line) or miR-155–deficient plasmablast B cells (black line) compared with plasmablast B cells from control mice not injected with EdU (grey) at 4 d post immunisation. Data are representative of at least two independent experiments.

also analysed using GOrilla. Some of the up-regulated processes involved regulation of cellular metabolic processes, mRNA splicing, as well as histone and chromatin modification (Fig 4D and Table S2).

The results in this report demonstrate a critical in vivo role for miR-155 in proliferation and survival of plasmablast B cells in response to the T-cell–dependent antigen HEL. We have used the $SW_{HEL}$ system to demonstrate that the impact of miR-155 deficiency becomes evident in B-cell blasts as early as 2.5 d after antigen stimulation, preceding differentiation into plasmablasts and also germinal centres. As B-cell blasts expand into extrafollicular plasmablasts, we show an essential role for miR-155 in sustaining the number of plasmablast B cells and in the production of class-switched antibodies. The data suggest that failure of miR-155

deficiency in sustaining the extrafollicular plasmablast response is due to impaired cell cycle progression and increased apoptosis. Indeed, it has been shown previously that B-cell differentiation and isotype switching in vitro requires cell division (Hodgkin et al, 1996) and miR-155–deficient splenic B cells cultured with LPS and IL-4 exhibited reduced frequency of CFSE$^+$ CD138$^+$ B cells compared with wild-type mice (Lu et al, 2014). At the peak of the plasmablast response, mechanistically miR-155 deficiency resulted in decreased mRNA abundance of genes involved in DNA replication, cell cycle progression, and chromatin organisation in an indirect manner. Some of these down-regulated genes included cell cycle regulators such as *E2F1*, *E2F2*, and *Myc*, which have been observed to affect B-cell differentiation in other systems (Lam et al, 1999; Shaffer et al,

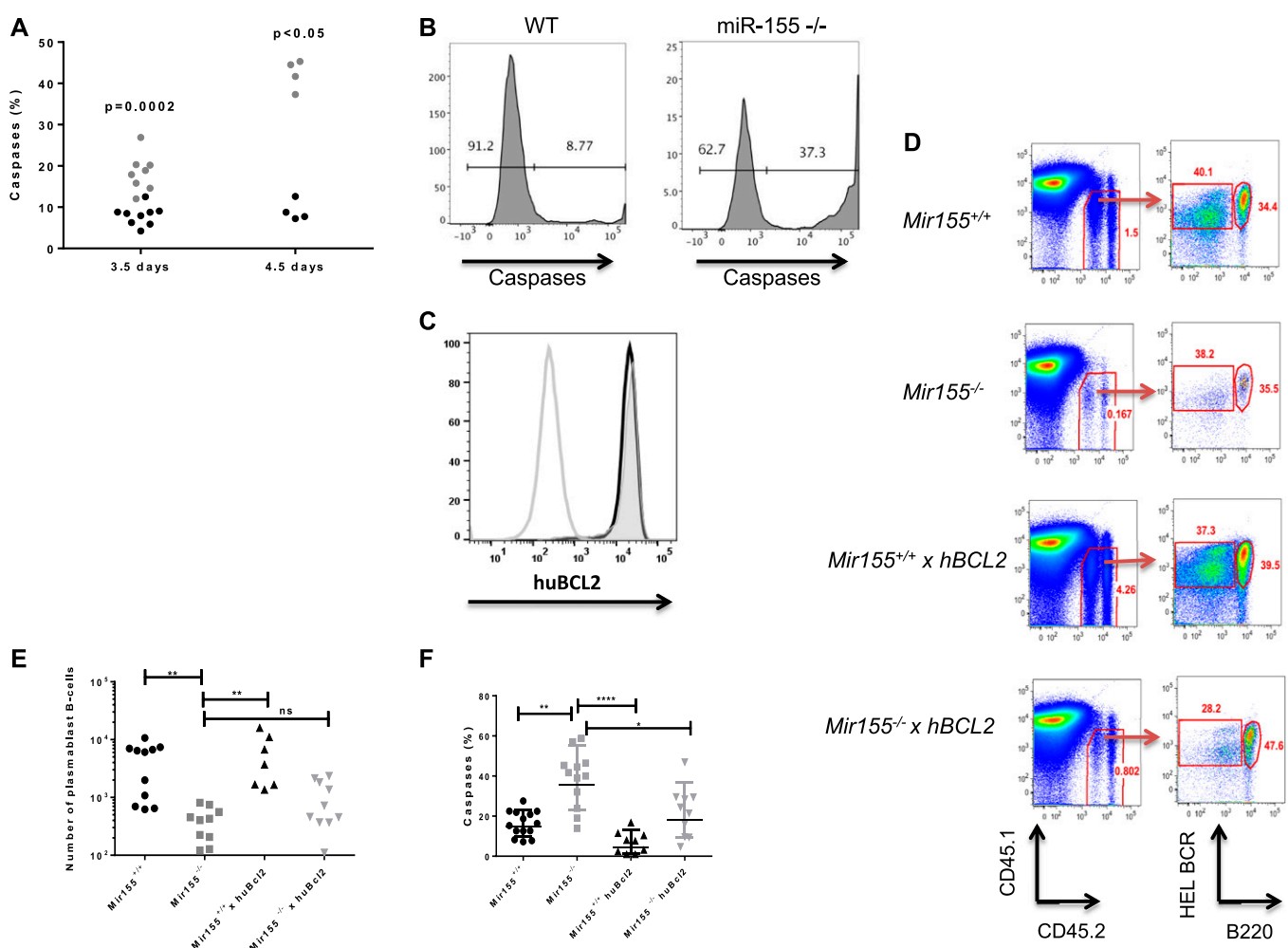

**Figure 3. miR-155 regulates survival of plasmablast B cells.**
**(A)** SW_{HEL} HEL BCR^+ B220^{lo} Mir155^{+/+} (black) or Mir155^{−/−} (grey) plasmablast B cells were analysed for activated caspases at the time points shown. A nonparametric Mann–Whitney test was used per time point. **(B)** Representative flow cytometry histograms of cleaved caspase detection in wild-type or miR-155^{−/−} SW_{HEL} plasmablast B cells at day 4.5 post immunisation. **(C)** Expression of the human Bcl2 transgene (huBCL2) in SW_{HEL} Mir155^{+/+} × Bcl2 (black line) or SW_{HEL} Mir155^{−/−} × Bcl2 (solid grey line) plasmablast B cells, compared with negative control (single grey line). **(D)** Representative FACS plot showing the gating strategy for analysis of adoptively transferred SW_{HEL} Mir155^{+/+} or Mir155^{−/−} B cells expressing a human Bcl2 transgene at day 4.5 post immunisation. Plots were previously gated on lymphocytes and single cells and analysed for CD45.2 donor cells and HEL^+ B220^{lo} plasmablast B cells. HEL^+ B220^{hi} germinal centre B cells could also be visualised in all mouse strains. **(E)** The number of splenic SW_{HEL} Mir155^{+/+} or Mir155^{−/−} plasmablast B cells in mice with or without the expression of a human Bcl2 transgene per 10^6 lymphocytes. A nonparametric Kruskal–Wallis with Dunn's multiple comparisons test was used where **P < 0.01. **(F)** The frequency of active caspases in miR-155–sufficient and miR-155–deficient plasmablast B cells at day 4.5 post immunisation. At least two independent experiments were carried out. Statistics calculated using nonparametric Kruksal–Wallis with Dunn's multiple comparisons test where *P < 0.05, **P < 0.01, ****P < 0.0001. Data are representative of at least two independent experiments.

2002). Previously, most of the roles of miR-155 in B-cell differentiation have been attributed to regulation of the germinal centre response, whereas the requirement for miR-155 in the extrafollicular plasmablast response has not been well characterised. Our data are significant in elucidating miR-155 as a new player in the early expansion of antigen-specific B-cell blasts into extrafollicular plasmablasts, which is necessary for short-term immune protection to infection (Nutt et al, 2015). The onset of the proliferation defect in Mir155^{−/−} B-cell blasts occur at the stage of B:T cell interaction, and we also previously showed that dysregulation of PU.1 by miR-155 in cultured B cells affects the expression of genes involved in adhesion and B:T cell interaction (Lu et al, 2014). However, SW_{HEL} Mir155^{−/−} mice from day 3.5 post HEL-SRBC

immunisation are still able to form germinal centres or locate in the red pulp (Nakagawa et al, 2016), suggesting that loss of cellularity in the absence of miR-155 is not explained by impaired migration and that, to some extent, miR-155–deficient B-cell blasts are responsive to T-cell help. The resemblance of our phenotype to that observed in IL-21 or IL-21R receptor–deficient mice (Lee et al, 2011), however, may indicate a potentially impaired response to T-cell help. In the absence of IL-21R, both the extrafollicular plasmablast and the germinal centre responses are impaired to ~10% of wild-type numbers (Lee et al, 2011), similar to what we have observed in miR-155–deficient mice. Furthermore, in one report, IL-21 signalling and miR-155 was reported to be linked in CD4^+ T cells from systemic lupus erythematosus patients (Rasmussen et al, 2015). It would,

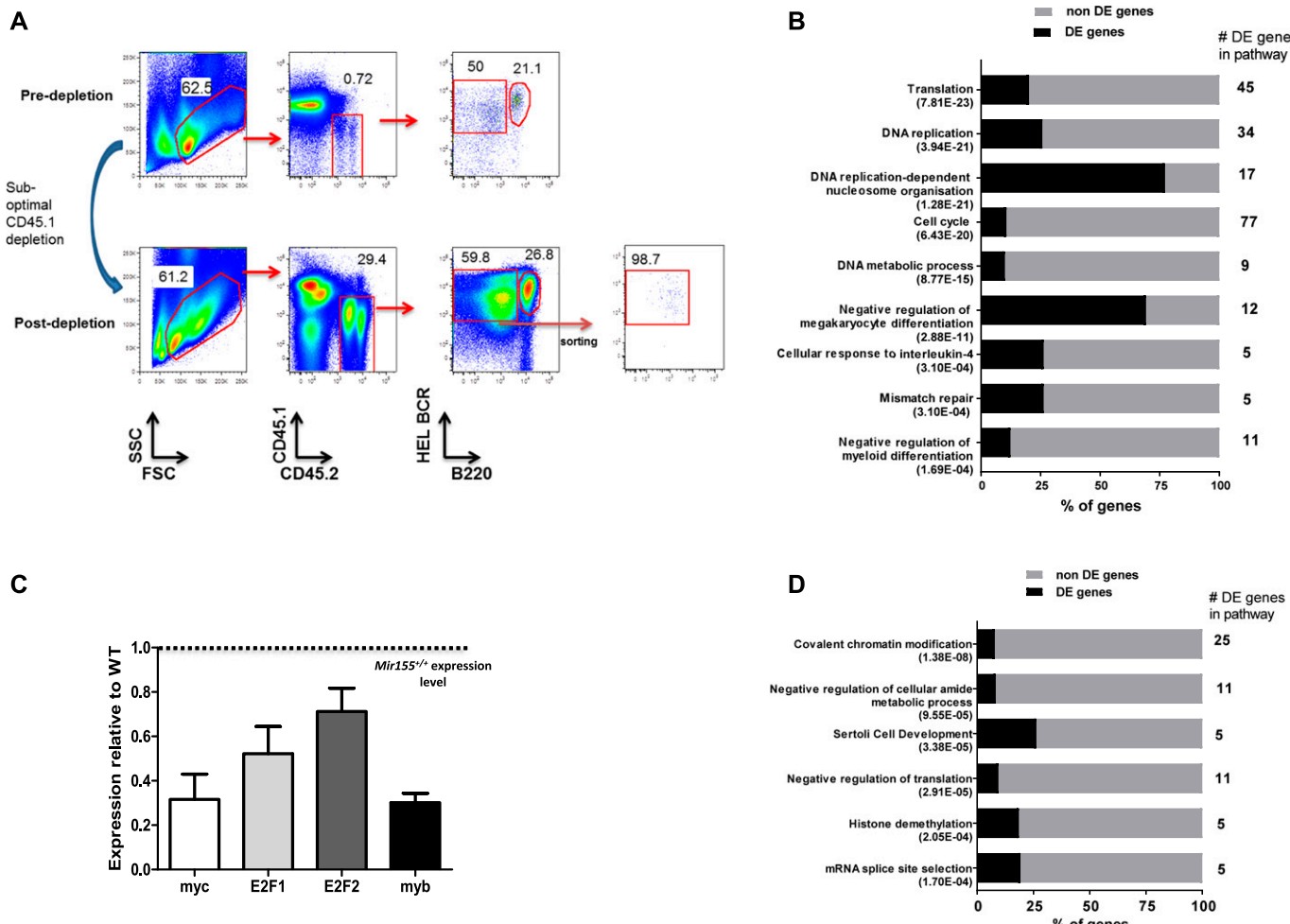

**Figure 4. miR-155 regulates the expression of genes associated with DNA replication.**
**(A)** Representative gating strategy showing CD45.2⁺ plasmablast B cells before and after suboptimal depletion of CD45.1⁺ cells. After enrichment, SW_HEL Mir155⁺/⁺ or Mir155⁻/⁻ plasmablast B cells were sorted on a FACS Aria to more than 98% purity and analysed by microarray analysis. N = 4–5 Mir155⁺/⁺ and 8–10 Mir155⁻/⁻ mice at day 4.5 post immunisation. **(B)** Differentially expressed (DE) genes with known functions were discovered by GOrilla analysis using the genes that were down-regulated in SW_HEL Mir155⁻/⁻ plasmablasts compared with SW_HEL Mir155⁺/⁺ plasmablasts. **(C)** The mRNA abundance of down-regulated genes with reported roles in DNA replication and function in Mir155⁻/⁻ plasmablasts relative to Mir155⁺/⁺ plasmablasts (dotted line) was confirmed by RT-qPCR. Expression values are normalized to HPRT, determined from three to four biological replicates from three to four independent sorting experiments using 4–5 Mir155⁺/⁺ or 8–10 Mir155⁻/⁻ mice per group. **(D)** GOrilla analysis of the DE genes that were up-regulated in SW_HEL Mir155⁻/⁻ plasmablast B cells compared with SW_HEL Mir155⁺/⁺ plasmablast B cells.

therefore, be interesting to investigate further whether the defects observed with miR-155 deficiency are linked to defective IL-21 signalling.

# Materials and Methods

## Mice

CD45.1⁺ congenic mice were bred and maintained in the Babraham Research Campus small animal facility. SW_HEL mice and miR-155–deficient mice and have been described previously (Phan et al, 2003; Rodriguez et al, 2007). SW_HEL mice were a gift from R. Brink (Garvan Institute of Medical Research/University of New South Wales). All mice were on the C57BL/6 background and bred and

maintained in the Biological Support Unit of Babraham Institute under specific opportunistic pathogen-free conditions.

## Adoptive transfer

SW_HEL Mir155⁺/⁺ or SW_HEL Mir155⁻/⁻ donor B cells were adoptively transferred into nonirradiated CD45.1⁺ congenic recipient mice followed by injection of 2 × 10⁸ HEL-SRBCs (Fig 1A). HEL (Sigma-Aldrich) was conjugated to SRBCs, and expression was subsequently measured with anti-HEL HyHEL9 antibody by flow cytometry. For proliferation experiments, donor B cells were labelled with 5 μM CFSE before adoptive transfer. For EdU incorporation studies, mice were injected with EdU i.p. 4 h before being euthanised. Spleen cells from mice were analysed by flow cytometry on the indicated days. For FACS analyses, CD45.2⁺ donor

splenocytes were enriched by CD45.1-negative selection using an autoMACS pro separator (Miltenyi Biotec).

### Flow cytometry

Multicolor flow cytometry for analysis or for sorting was performed on an LSR Fortessa-5 or FACS Aria (BD Biosciences), respectively. Single-cell suspensions of splenocytes were blocked with anti-CD16/32 mAb (clone 2.4G2), followed by staining with the following antibodies: anti-B220 (clone RA3-6B2) and anti-CD45.2 (clone 104) from BD Biosciences. HEL-binding B cells were stained as described previously (Chan et al, 2009). For cell cycle analyses, spleen cells were first stained for extracellular antigens and then were analysed with 10 $\mu$g/ml DAPI staining using a Cytofix/Cytoperm kit (BD Biosciences) or PFA and Tween-20. Cell cycle was calculated by FlowJo Dean/Jett/Fox algorithm or by setting gates manually. The Click-iT EdU Alexa Fluor 488 Imaging kit and CaspGLOW Fluorescein Active Caspase Staining kit (both from Thermo Fisher Scientific) were used according to the manufacturer's instructions. Data were analysed with FlowJo software (Tree Star).

### Microarray

After sorting of plasmablast B cells directly into Trizol, RNA was extracted and resuspended in RNAse-free water. RNA that passed quality control using a bioanalyzer and NanoDrop was subject to rounds of amplification using the Ambion Expression kit. cDNA from five independent biological samples of SW$_{HEL}$ *Mir155*$^{+/+}$– or SW$_{HEL}$ *Mir155*$^{-/-}$–sorted plasmablast B cells were hybridized to GeneChip Mouse Gene ST1.0 arrays (Affymetrix) according to the manufacturer's instructions. Bioconductor package affy and the robust multiarray average function were used for background correction, and normalization was performed using the software package R by the Babraham Bioinformatics facility. Normalized data were filtered with a threshold of the modal expression value in which three of the five samples had to exceed the log2 modal expression threshold. Differentially expressed genes between miR-155–sufficient or miR-155–deficient plasmablast B cells were assessed using a *P*-value less than 0.05 and a fold change of greater than 1.3-fold.

### Gene ontology analysis

Differentially expressed genes in miR-155–deficient plasmablast samples were computed using the GOrilla tool (Eden et al, 2009) to determine enriched gene ontology terms. A background list of genes was included in the analysis. If several related terms were significantly enriched, the terms with a higher percentage of differentially expressed genes were chosen, and are presented in Fig 4.

### DNA isolation, RNA extraction, and RT-qPCR assays

Total RNA was extracted from sorted plasmablast populations using TRIzol (LifeTech). RNA from sorted plasmablast B cells was converted to cDNA according to the superscript reverse transcriptase III

protocol (Invitrogen) and then analysed by RT-qPCR. Cell cycle genes E2F1, E2F2, Myc, and Myb were analysed using custom or commercially available primers (see Table S3). E2F1 and E2F2 mRNA transcript expression was analysed using primers according to Pilon et al (2008). Myc, E2F1, and E2F2 RT-qPCR assays were analysed using Platinum SYBR Green qPCR SuperMix (Life Technologies). Relative abundance was calculated using a standard curve or $\delta$ CT method and normalized to the expression of mRNA-encoding HPRT. Myb RT-qPCR assays were performed with Taqman assays. Expression of Myb mRNA was calculated using a standard curve and normalized to the expression of $\beta$2M.

### Statistics

Statistical analyses were performed in GraphPad Prism software or R Studio; tests are indicated in the figure legends. All data were tested for normality of residuals. If data were normally distributed, parametric tests were used. For non-normally distributed data, several transformations were attempted, and if this was satisfactory, parametric tests were used. Where transformation of data yielded non-normally distributed residuals, nonparametric tests were used. For B-cell blasts number, data were square-root transformed. For testing the effects of genotypes on plasmablast and germinal centre cell number, the data were log-transformed and tested by two-way ANOVA. There was a significant interaction between genotype and time. For CFSE data, the values were arcsine-transformed. For EdU incorporation data, the values were arcsine-transformed and tested by two-way ANOVA.

### Study approval

All mouse experiments were approved by the Animal Welfare and Ethical Review Body of the Babraham Institute. Animal husbandry and experimentation complied with the existing European Union, United Kingdom Home Office legislation and local standards.

## Supplementary Information

## Acknowledgements

We wish to thank the Biological Services Unit, Flow Cytometry core, and Bioinformatics facilities of the Babraham Institute for expert technical assistance; Dr Robert Brink at the Garvan Institute of Medical Research, Australia, for kindly providing the SW$_{HEL}$ mice; the Core Genomics Lab at Addenbrooke's hospital for RNA quality control, amplification, and hybridisation; Dr Lucy Crooks at the Biomolecular Sciences Research Centre for providing statistical input; Dr Sarah Bell at the Babraham Institute for providing scientific input; and other members of the Vigorito and Turner lab for helpful discussions. This work was supported by Biotechnology and Biological Sciences Research Council and Medical Research Council grants BB/J00152X/1 and BBS/E/B/000C0407.

## Author Contributions

G Arbore: data curation, formal analysis, investigation, and methodology.

T Henley: data curation, formal analysis, investigation, and methodology.

L Biggins: formal analysis and investigation.

S Andrews: formal analysis and investigation.

E Vigorito: conceptualization, resources, supervision, funding acquisition, and writing—original draft, review, and editing.

M Turner: funding acquisition and writing—review and editing.

R Leyland: resources, data curation, formal analysis, supervision, investigation, methodology, project administration, and writing—original draft, review, and editing.

## Conflict of Interest Statement

The authors declare that they have no conflict of interest.

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
