## [Reviewer comments · Life Science Alliance]

Life Science Alliance

MicroRNA-155 is essential for the optimal proliferation and survival of plasmablast B cells

Giuseppina Arbore, Tom Henley, Laura Biggins, Simon Andrews, Elena Vigorito, Martin Turner, and Rebecca Leyland

DOI: <https://doi.org/10.26508/lsa.201800244>

Corresponding author(s): Rebecca Leyland, Sheffield Hallam University

Review Timeline:

Submission Date:	2018-11-14
Editorial Decision:	2018-12-20
Revision Received:	2019-03-26
Editorial Decision:	2019-04-12
Revision Received:	2019-04-29
Accepted:	2019-04-30

Scientific Editor: Andrea Leibfried

Transaction Report:

December 20, 2018

Re: Life Science Alliance manuscript #LSA-2018-00244-T

Dr. Rebecca Leyland
Babraham Institute
Cambridge
United Kingdom

Dear Dr. Leyland,

Thank you for submitting your manuscript entitled "MicroRNA-155 is essential for the optimal proliferation and survival of plasmablast B cells" to Life Science Alliance. The manuscript was assessed by expert reviewers, whose comments are appended to this letter.

As you will see, the reviewers raise similar concerns regarding data representation, robustness of the analyses, lack of statistics and lack of quantifications. Reviewer #2 furthermore thinks that more insight would be needed at this stage. If you can address the technical issues / data presentation issues, we'd be happy to invite you to submit a revised version of this work. Importantly, except for reviewer #2's request for further reaching insight, which is not necessarily needed to allow publication here, all other concerns need to get addressed fully. Please note that we would need strong support from the reviewers on such a revised version.

Thank you for this interesting contribution to Life Science Alliance. We are looking forward to receiving your revised manuscript.

Sincerely,

- A letter addressing the reviewers' comments point by point.
- An editable version of the final text (.DOC or .DOCX) is needed for copyediting (no PDFs).
- High-resolution figure, supplementary figure and video files uploaded as individual files: See our detailed guidelines for preparing your production-ready images, <http://life-science-alliance.org/authorguide>
- Summary blurb (enter in submission system): A short text summarizing in a single sentence the study (max. 200 characters including spaces). This text is used in conjunction with the titles of papers, hence should be informative and complementary to the title and running title. It should describe the context and significance of the findings for a general readership; it should be written in the present tense and refer to the work in the third person. Author names should not be mentioned.

B. MANUSCRIPT ORGANIZATION AND FORMATTING:

Full guidelines are available on our Instructions for Authors page, <http://life-science-alliance.org/authorguide>

Reviewer #1 (Comments to the Authors (Required)):

The manuscript "microRNA-155 is essential for the optimal proliferation and survival of plasmablast B cells" by Leyland et al. explores further the role of miR-155 in the humoral immune response, expanding previous studies by the same authors. In this manuscript the authors focus on the early

extrafollicular immune response and the generation of plasmablasts using an elegant combination of miR-155 deficiency with a well-characterized antigen specific model (SW-HEL). The authors conclude that miR-155 regulates plasmablast generation in response to antigenic exposure impinging on both the proliferation and survival of the cells. The claims are interesting and well-sustained by the data, but some clarifications would be advisable, as detailed below:

Figure 1. miR-155 is required for plasmablast response.

- 1) Please clarify Figure 1A legend. How was HEL expression measured? What was used as control cells?
- 2) In Figure 1B it is really hard to see the differently colored cell subsets. Showing individual FACS plots for each subset would be very useful, for instance, a HEL/CD138 plot after B220 gating, etc. Enlarging the plot would also help. Please define B-blasts in text. Could the authors show some "specific" germinal center marker, such as FasvsGL7 or PNA?
- 3) In Figure 1B, and related to previous, authors state that all B-cell blasts, plasmablasts and germinal center B cells are reduced in the absence of miR-155 (p5), please show quantification and statistics. One would assume that germinal center B cells would be very scarce at this time point?
- 4) For the sake of validating the antigen-specific model in combination with miR-155 deficiency, it would be nice to replicate the germinal center defect at later timepoints, as shown in the Vigorito et al 2007 paper.
- 5) Please provide statistics for Figure 1D.

Figure 2. miR-155 deficiency affects proliferation.

- 1) Figure 2A shows a proliferation defect in miR-155 deficient cells by CFSE staining dilution. To convey this message clearly, plots should be cleaner, i.e. thinner lines to visualize the division peaks, and quantification with statistics provided. Also, at days 2.5 and 3.5 plots show a very bright peak (expectably, undivided cells) that is not present at earlier time points, what does it mean?
- 2) Figure 2B and C. Please provide representative plots Edu/DAPI (or similar).
- 3) Figure 2B. Authors show percent of Edu+ cells as a readout for DNA incorporation at day 4 and at day 4.5. The difference in Edu incorporation observed in this 12 hour lapse is very impressive (it drops from roughly 45% Edu+ at d4 to less than 10% at d4.5), could the authors discuss this point? It would be very useful to also see Edu incorporation at earlier timepoints, similar to the kinetics performed elsewhere in the manuscript. Likewise, could the authors analyze the proportion of G1 and S/G2/M at earlier timepoints?

Figure 3. miR-155 deficiency affects survival.

- 1) Figure 3A. Please show representative FACS plot of Caspase staining.
- 2) Figure 3C. Please enlarge plots and font.
- 3) Figure 3D. The effect of Bcl2TG is somehow unclear in this graph. Significant differences are indicated for miR155+/+ vs miR155-/- (as shown before), and also between miR155+/+Bcl2TG and miR155-/-Bcl2TG. However, the rescue of miR155 by Bcl2 (miR155-/- vs miR155-/-Bcl2TG) seems very minor and apparently not significant. Likewise, the observation "the rescue of miR155-/- plasmablasts was greater than in miR155+/+ mice" (p7), is also unclear.
- 4) Figure 3E. The data are very convincing that Bcl2 decreases apoptosis in both miR155+/+ and MiR155-/- backgrounds (expectedly). This is however overlapping with the data shown in 3A (where the result is already shown for miR155+/+ and miR155-/-). Maybe both sets of data could be consolidated as 3A
- 5) In the light of the results in Figure 3D and 3E, the authors conclude that as Bcl2 expression clearly decreases apoptosis (3E) but alone does not have a major role on plasmablast cell number (3D), mechanisms other than apoptosis also play a role (p7). Presumably this mechanism is related with replication/proliferation (as shown in Fig 2). Therefore, it would be nice to see that Bcl2 TG

indeed does not overcome the proliferation defect in miR155 deficient B cells.

Figure 4. Transcriptome analysis.

1) Figure 4A. Please enlarge plots and font.

2) The transcriptome analysis can be complemented with a target analysis enrichment for downregulated genes.

General comment. This manuscript, together with previous reports by the same authors and other labs, highlight the pleiotropic effect of miR155 during the immune response. Even within the B cell autonomous functions, the impact of miR155 is rich and entangled. It would be very useful if the authors could make some comparisons or integrative summary/cartoon of the targets and different pathways regulated by miR-155 at different stages of the B cell response (most notably, short lived plasmablasts, germinal center, plasma cell and memory B cells).

Reviewer #2 (Comments to the Authors (Required)):

This manuscript submitted to LSA focusses on the role of a well-documented regulatory microRNA cluster, mir155, on its roles in the early plasma blast response. The study develops on the authors' previous work to understand the nature of a PB genesis defect in mir155^{-/-} mice. The body of work uses Rob Brink's SwHEL B cell system, adoptively transferring congenic naive CD45.2⁺ mir155^{-/-} and ^{+/+} B cells into Ly5.1 mice with concomitant HEL-SRBC immunisation. In the days preceding GC coalescence, the authors examine the division, differentiation and transcriptomic profile of the developing B cell and PB response. They find differences in the expansion of ^{-/-} SwHEL B cells and PB in terms of number and progression through multiple division rounds, evidence of increased apoptosis and less frequent S-phase entrance in the ^{-/-} cells. The transcriptomic profile fits with a role for mir155 in regulating numerous cycle-associated genes. The overall conclusion is that by both limiting apoptosis and supporting cell cycle progression, mir155 contributes to the continuance of the extra follicular PB response.

In general, the data support the primary conclusion. However, I would consider this a preliminary version of the manuscript that is not yet ready for publication. The conclusions are somewhat overstated. Key data points are not strong enough to draw robust conclusions. To strengthen this manuscript for reevaluation, I would request:

1B, - A more complete phenotypic description of these populations, at least in the wt SwHEL cells, along with a more complete gating hierarchy should be included.

1D, - Are these from three mice in one experiment, or pooled sera from three experiments? These data are quite important for the studies' focus and I would suggest should be analysed independently at least twice.

2A, - The data are supported, but are shown from a single mouse. Some indication of variation in multiple mice would help support the concluded lack-of-difference in the day 1.5 timepoint vs the established change on days 2.5 onwards.

3D - The key conclusion discussed in the results is based on what appears to be a statistically non-significant difference. Also, are these total SwHEL B cells or PB? The results discuss PB but the legend references B cells. Exact #s of PB is relevant in this case. This is important as IRF4, a key PC fate-programming gene is affected by PU.1 which the authors have shown previously to be affected by mir155. A rescue of B cells could feed on to enhance PB #s, but this would not necessarily be PB-intrinsic, even with the caspase and Edu/DAPI profiles shown.

In addition to strengthening these current data and tightening the manuscript I would request two additional experiments or justifications. The argument that there is an intrinsic role for mir155 in the PB response, while supported by the data, do not indicate it is PB-intrinsic but rather B cell-intrinsic. An effect on the ability of B cells to activate the perfollicular T cells that support the preGC response (and give rise to PB) would equally feed into a numeric PB phenotype defect. To show a direct intrinsic role, I would suggest that placing the mir155^{-/-} SwHEL cells into competition with ^{+/+} SwHEL cells by co-transfer to eliminate feedback elements that could arise if at the B cell-level, mir155 is impacting the activation of interacting T cells prior to PB differentiation. In this co-transfer context the feedback defects will not compound, such that T cells are equally primed, but defects more directly intrinsic and dependent on the ability of the ^{-/-} B cells to receive T cell help. This could be evaluated by co-transferring CD45.1/2 and CD45.2/2 ^{+/+} and ^{-/-} cells into CD45.1/1 mice that I think could be generated in a 1 or 2 generation cross depending on the SwHEL breeding setup. For the early time points (d1.5/2.5), using two different proliferation dyes (CFSE vs CTV) could reach the same point without any necessary intercrosses.

A second argument is made that it is only in the window at which T cell help is beginning that proliferative differences begin. In addition to more data points for figure 2A, an experiment that would further support this finding would be to demonstrate (or cite) that anti-IgM drives similar proliferative responses in ^{+/+} and ^{-/-} B cells in this window, or test whether non-SwHEL miR155^{-/-} and ^{+/+} NP-Ficoll responses or LPS PB responses are equivalent or not.

-- Additional points for clarification:

The text needs to be tightened. There are numerous instances throughout in which the results section are disconnected from the figure legends (for example, the results mentions a green group in Figure 1B, but these cells are grey in the figure and associated legend). The methods section lacks some details, such as how EdU was administered to the mice. The gating strategies used, while previously published by other groups, are a little too ambiguous to be fairly evaluated in this study. Specifically, GC B cells are typically identified by FAS, GL7, PNA, Ephrin B1 or BCL6 expression, along with downmodulation of CD38 and IgD; other strategies are possible and often appropriate, such as the putative strategy employed here, being down modulation of BCR relative

to activated non-GC B cells, but as far as I can ascertain, the low HEL expression cited to identify GC B cells is actually referencing cytoplasmic HEL staining differences in PB and GC B cells. It is a little confusing to use surface BCR alone in CD138negative cells to distinguish preGC and GC B cells. Also, GC cells are essentially irrelevant for this manuscript, whereas clear demarcation of the PB is absolutely crucial. That is, stronger evidence that the distinction between B-blasts and committed GC cells is necessary, if these cells are to be discussed in the results section as in the current version. A phenotypic profile of the analysed populations would help.

With the exception of the seemingly appropriate GO expt execution and analysis (Figure 4), many of the experiments that seem well executed lack statistical rigour (Figure 1-3) which makes it difficult to have confidence in the strength of the associated conclusions. There are places in the results section where I view the conclusions as overstated based on the presented data. That is, where strong conclusions are made, the analysed data are not significant at that point (esp. 3C&3D); the reasons such inferences are suitable given the outcome needs to be more clearly justified, or the presented data reconsidered to address the major conclusion that is intended from the experiment. In numerous cases, multiple time points are analysed. The authors have, appropriately, selected non-parametric analyses, yet employ (inappropriately, and without suitable justification) multiple T-tests on these data. Post-hoc analyses following non-parametric anova are more typical analyses for the field. Also, some data are presented on log-axes, yet averages are shown as arithmetic mean + SEM, which provides little information about the distribution of a log-normal population. More appropriate is either geometric mean + geo SD factor or median + IQR. Where non-standard analyses are employed, I request that the authors justify these uses more explicitly. In some such cases, parametric analyses after log-transformation can also be appropriate in place of non-parametric tests. For example, for Figure 1C, the difference is reasonably clear and the expt seems sufficiently powered to reveal the argued difference(s), but a T-test is an atypical and I suggest, an inappropriate test to employ in such a case.

The claim is made from Fig 2 that by having more cells in G1 and fewer in S/G2/M (Figure 2C), mir155 is impacting progression from G1 to S. However, DAPI staining does not distinguish G1 from G0 cells, and every aspect of the study could be ostensibly explained by a larger fraction of cells in each division becoming quiescent. The data could equally indicate the cells have a reduced division destiny or receive poorer T cell help and are thus prone to quiescence. Also the HEL-SRBC activation system essentially activates the BCR concomitant with transfer, which could induce proliferation before receipt of T cell help. To claim that the loss of proliferation at day 2.5 is related to T cell help, these data would be strengthened by showing more than one representative mouse for each timepoint (Figure 2A), or some in vitro anti-BCR studies in mir155^{-/-} and ^{+/+} B cells, or reference to such studies.

The data in 3C & 3D are used to argue that PB are more numerous and frequent in mir155^{-/-}-BCL2-rescued cells. 3C actually shows a proportional reduction in PB in mir155^{-/-}-BCL2 vs mir155^{-/-}, while 3D shows a statistically non-significant increase in (from the legend) B cell numbers. It is further unclear from the text+figure+legend whether these encompass B cells + PB, just B cells, or just PB.

While these can be used to back-calculate the # of PB, this is a rather circuitous way to get to the argument that (I think) the authors are trying to make. The key parameter (#of PB) needs to be evaluated directly, the power of the experiment increased, or the experiment redesigned to address the question as discussed and interpreted in the results section. As this is a key argument for the paper, I think the conclusion is overstated.

Minor points:

- In the results/discussion, please cover more explicitly and appropriately reference that the role for miR155 in PB response has been shown rather than is established here, but these studies are confirming the phenotype in SwHEL system.

In the discussion of Figure 1B the authors refer to plasma blast B-cells, and then to extra follicular PB in 1C. I would suggest they reconsider their definition of a PB to the more conventional proliferating cell producing antibody, whereas a B-blast is a proliferating focus of B cells. Making these terms more explicit would help clarify the text.

1D please state what each connected line represents (i.e., individual mice). Was 1D repeated in an independent experiment? 1C Where it says 'data is representative of at least two independent experiments' please state the actual number of experiments used to generate data. Also, I think this should probably read data were combined from two experiments, rather than representative of one of two.

2A - It would be nice to see what % of the HEL B cells in each case had divided at all (I can't see it in the histogram)

2B please show representative flow plots of Edu staining on the PB

2C please show representative flow plots of DAPI staining on the PB. Stats: These data are paired (ie., repeated measures).

2A-C please state # of expts

The manner in which EdU was administered is not stated, and if it was administered i.p., the time after administration that euthanasia was performed should be stated.

In the discussion of figure 2C, I suggest the conclusion overstates the finding. The assays used do not distinguish between G0 and G1 phases; to state that G1/S phase transition is blocked is inappropriate. The conclusion that fewer cells are now in S/G2/M is, however, appropriate.

3A please show representative flow plots

3C - showing the individual experimental results would be helpful. Otherwise please state what the average and error bars show. Also please consider if the type of error bar shown is appropriate for DDCT data.

3D please state what error bars represent. Please do not use SEM

3D these data are difficult to interpret. The most meaningful c.f. would be g2 vs g4, but these are not different. But there is a 2-3-fold shift in average (is it mean here?), which is hard to argue is meaningless, but rather underpowered. This seems to be a key comparison for the study, but I would suggest the strength of the conclusion is overstating the strength of these data. As presented, these data show more (but not statistically significantly more) B cells, but relative to each B cell, a smaller fraction of PB (3C). If the argument is about PB, please use data that more overtly demonstrates the point.

Also, if all data points are shown, it is not representative, but rather all-encompassing, so please be clear with what your data are reflecting.

3E - these differences are to be expected, yet, there seems a limit on absolute possible number of PB in the wt: if fewer are dying but # is similar, what is happening to them? i.e., did you look at bone marrow for early PC seeding?

In the conclusions to Figure 3, they discuss that in addition to rescuing PB from apoptosis, other mechanisms come into play. BCL2 also rescues B cells from apoptosis, so how is it distinguished that this rescue relates to anti-apoptotic effects at the PB level rather than simply as a function of having more B cells available for differentiation?

Reviewer #3 (Comments to the Authors (Required)):

This manuscript from Leyland et al., appears to be an interesting story about the contribution of mir155 to plasmablast survival and proliferation. However, the authors have been quite careless in the process of putting their data together, making assessment a little difficult. I have got the gist of the story nonetheless and will likely be positive if some attention is paid to making the data clearer as pointed out below:

On page 5 the authors state that plasmablasts, defined as being B220^{lo} HEL BCR⁺ and CD138⁺, were represented in green in Figure 1B. In Figure 1B there is no indication of the CD138 gate and the plasmablasts are hard to distinguish with the colors used. They are in "grey" not "green" but this figure needs to be made clearer.

In Figure 2 we are apparently looking at CFSE dilution in plasmablasts but how these cells were identified is not clear in the text, the legend or most importantly in the figure itself.

In Figure 3 it appears that CD138 is no longer used to gate on plasmablasts -at least not based on the statement in the legend.

In figure 3C the CD45.2 gate is clear. No arrows to indicate that the right side panels are derived from the CD45.2 box. Assuming that this is so, the B220 low box on the right presumably represents plasmablasts but there is no indication provided in the legend or the figure as to what is in each box in this figure. What cells occupy the oval demarcations on the right of this box in Fig 3C is also not made clear.

Reviewer #1 (Comments to the Authors (Required)):

The manuscript "microRNA-155 is essential for the optimal proliferation and survival of plasmablast B cells" by Leyland et al. explores further the role of miR-155 in the humoral immune response, expanding previous studies by the same authors. In this manuscript the authors focus on the early extrafollicular immune response and the generation of plasmablasts using an elegant combination of miR-155 deficiency with a well-characterized antigen specific model (SW-HEL). The authors conclude that miR-155 regulates plasmablast generation in response to antigenic exposure impinging on both the proliferation and survival of the cells. The claims are interesting and well-sustained by the data, but some clarifications would be advisable, as detailed below:

Figure 1. miR-155 is required for plasmablast response.

1) Please clarify Figure 1A legend. How was HEL expression measured? What was used as control cells?

We thank the reviewer for highlighting this. HEL expression was measured by conjugation of HEL to SRBCs and subsequent detection using anti-HyHEL9-APC antibody. This information has now been added to the methods section and the legend for Figure 1A.

2) In Figure 1B it is really hard to see the differently colored cell subsets. Showing individual FACS plots for each subset would be very useful, for instance, a HEL/CD138 plot after B220 gating, etc. Enlarging the plot would also help. Please define B-blasts in text. Could the authors show some "specific" germinal center marker, such as FasvsGL7 or PNA?

Figure 1B now shows a gating strategy with individual FACS plots. After HEL BCR/B220 gating, plasmablast B-cells are defined with the additional marker CD138. Germinal centre B-cells are defined further using the marker FAS/CD95, as suggested by the reviewer. B-cell blasts are defined within the text.

3) In Figure 1B, and related to previous, authors state that all B-cell blasts, plasmablasts and germinal center B cells are reduced in the absence of miR-155 (p5), please show quantification and statistics. One would assume that germinal center B cells would be very scarce at this time point?

The quantification of B-cell blasts, plasmablasts and germinal centre B-cells are now added as Figure 1C with statistics included.

4) For the sake of validating the antigen-specific model in combination with miR-155 deficiency, it would be nice to replicate the germinal center defect at later timepoints, as shown in the Vigorito et al 2007 paper.

We did not assess the kinetics of the germinal centre response at later time points for these experiments, as the focus of the project was to elucidate the role of miR-155 in the extrafollicular plasmablast response.

5) Please provide statistics for Figure 1D.

We have carried out statistics for WT and miR-155^{-/-} values at all dilutions, whereby all isotype differences can be captured and used a Two Way ANOVA with Sidak's multiple comparison test. This is now included in Figure 1D and p values indicated in the figure legend.

Figure 2. miR-155 deficiency affects proliferation.

1) Figure 2A shows a proliferation defect in miR-155 deficient cells by CFSE staining dilution. To convey this message clearly, plots should be cleaner, i.e. thinner lines to visualize the division peaks, and quantification with statistics provided. Also, at days 2.5 and 3.5 plots show a very bright peak (expectably, undivided cells) that is not present at earlier time points, what does it mean?

CFSE proliferation plots in Figure 2A have now been revised according to the request of the reviewer. The lines are now thinner and clearer to allow better visualisation of the dividing peaks. The percentage of dividing cells in each generation has been quantified for days 2.5 and 3.5, with statistical analysis included (Figure 2B). The undivided control peak which was present on plots representative of days 2.5 and 3.5 has now been removed to avoid confusion in this figure, as raised by the reviewer, as this information is now quantified in Figure 2B.

2) Figure 2B and C. Please provide representative plots Edu/DAPI (or similar).

Representative flow cytometry plots for cell cycle analysis and EdU staining in both miR-155 sufficient and deficient SW_{HEL} plasmablast B-cells has now been added to Figure 2 and additional information is complete in the respective figure legend. These are now shown as Figures 2D and 2F.

3) Figure 2B. Authors show percent of Edu+ cells as a readout for DNA incorporation at day 4 and at day 4.5. The difference in Edu incorporation observed in this 12 hour lapse is very impressive (it drops from roughly 45% Edu+ at d4 to less than 10% at d4.5), could the authors discuss this point? It would be very useful to also see Edu incorporation at earlier timepoints, similar to the kinetics performed elsewhere in the manuscript. Likewise, could the authors analyze the proportion of G1 and S/G2/M at earlier timepoints?

Earlier time points for both cell cycle analysis and EdU incorporation have now been added to the manuscript as requested by the reviewer. For cell cycle analysis, we now show data at 3.5- and 4.5-days post immunisation in Figure 2C, with statistical analysis included and a representative flow cytometry plot in Figure 2D. Additional kinetic analysis of EdU at earlier time points is also now included as Figure 2E with an additional representative flow cytometry histogram to compliment this data in Figure 2F.

Figure 3. miR-155 deficiency affects survival.

1) Figure 3A. Please show representative FACS plot of Caspase staining.

Representative flow cytometry histograms to measure cleaved caspases in wild type or miR-155 deficient SW_{HEL} plasmablast B-cells have now been added to the manuscript. This is now presented as Figure 3B, with additional information added to the figure legend.

2) Figure 3C. Please enlarge plots and font.

As requested by the reviewer, the plots and fonts of the flow cytometric plots have now been enlarged and the font size increased. We hope this is to the reviewer's satisfaction.

3) Figure 3D. The effect of Bcl2TG is somehow unclear in this graph. Significant differences are indicated for miR155+/+ vs miR155-/- (as shown before), and also between miR155+/+Bcl2TG and miR155-/-Bcl2TG. However, the rescue of miR155 by Bcl2 (miR155-/- vs miR155-/-Bcl2TG) seems very minor and apparently not significant. Likewise, the observation "the rescue of miR155-/- plasmablasts was greater than in miR155+/+ mice" (p7), is also unclear.

We agree this data is an important part of the manuscript and we have therefore added further information within the text to clarify the extent of the increase observed in both SWHEL miR-155+/+ and miR-155-/- mice in the presence of the Bcl2 transgene. The following text has been added:

" The mean number of SW_{HEL} Mir-155^{+/+} plasmablast B-cells increased 1.3 fold in the presence of the Bcl2 transgene, whereas the number of SW_{HEL} Mir-155^{-/-} plasmablast B-cells increased 2.5 fold in the presence of the Bcl2 transgene, further supporting a role for miR-155 in maintenance of plasmablast survival. "

4) Figure 3E. The data are very convincing that Bcl2 decreases apoptosis in both miR155+/+ and miR155-/- backgrounds (expectedly). This is however overlapping with the data shown in 3A (where the result is already shown for miR155+/+ and miR155-/-). Maybe both sets of data could be consolidated as 3A

The Bcl2 transgene data was separated into two figures to help with the narrative of the manuscript and to illustrate different scientific conclusions. Figure 3A shows the difference in cleaved caspase expression between day 3.5 and day 4.5 post immunisation in plasmablast B-cells to emphasise the increase in apoptosis between these time points in the absence of miR-155. Figure 3E (now Figure 3F) compares the frequency of cleaved caspases in different mouse strains including with or without the Bcl2 transgene, in independent experiments for day 4.5 only. Both graphs include data from independent experiments.

5) In the light of the results in Figure 3D and 3E, the authors conclude that as Bcl2 expression clearly decreases apoptosis (3E) but alone does not have a major role on plasmablast cell number (3D), mechanisms other than apoptosis also play a role (p7). Presumably this mechanism is related with replication/proliferation (as shown in Fig 2). Therefore, it would be nice to see that Bcl2 TG indeed does not overcome the proliferation defect in miR155 deficient B cells.

We utilised the Bcl2 transgenic mice to assess the contribution of apoptosis to the phenotype in miR-155 deficient plasmablasts and found that the phenotype was partially rescued but the plasmablast cell number did not restore to wildtype levels. Due to the phenotype not being fully restored, we did not feel it was necessary to carry out proliferation assays in these mice.

Figure 4. Transcriptome analysis.

1) Figure 4A. Please enlarge plots and font.

As per the reviewers request, the FACS plot has now been enlarged.

2) The transcriptome analysis can be complemented with a target analysis enrichment for downregulated genes.

We used target scan software to try to identify direct miR-155 targets in both the upregulated and downregulated gene sets. We did not find any particular enrichment for miR-155 targets in either data set at the time point analysed. We also carried out miR-155 seed enrichment analysis using microRNA.org and we found that only 42 genes were found to have a highly conserved miR-155 binding site in the downregulated gene set and 78 genes were found to have a highly conserved miR-155 binding site in the upregulated genes. Therefore, there did not seem to be a particular enrichment of miR-155 targets.

General comment. This manuscript, together with previous reports by the same authors and other labs, highlight the pleiotropic effect of miR155 during the immune response. Even within the B cell autonomous functions, the impact of miR155 is rich and entangled. It would be very useful if the authors could make some comparisons or integrative summary/cartoon of the targets and different pathways regulated by miR-155 at different stages of the B cell response (most notably, short lived plasmablasts, germinal centre, plasma cell and memory B cells).

We thank the reviewer for this suggestion. However, within this manuscript we did not identify any additional targets of miR-155, as we focused on the cellular mechanisms of plasmablast regulation by miR-155. As the focus of this manuscript is on plasmablasts predominantly, a cartoon illustrating the role of miR-155 throughout the course of B-cell differentiation may have additional value in a review and this is something we will consider.

Reviewer #2 (Comments to the Authors (Required)):

This manuscript submitted to LSA focusses on the role of a well-documented regulatory microRNA cluster, mir155, on its roles in the early plasma blast response. The study develops on the authors' previous work to understand the nature of a PB genesis defect in mir155^{-/-} mice. The body of work uses Rob Brink's SwHEL B cell system, adoptively transferring congenic naive CD45.2⁺ mir155^{-/-} and ^{+/+} B cells into Ly5.1 mice with concomitant HEL-SRBC immunisation. In the days preceding GC coalescence, the authors examine the division, differentiation and transcriptomic profile of the developing B cell and PB response. They find differences in the expansion of ^{-/-} SwHEL B cells and PB in terms of number and progression through multiple division rounds, evidence of increased apoptosis and less frequent S-phase entrance in the ^{-/-} cells. The transcriptomic profile fits with a role for mir155 in regulating numerous cycle-associated genes. The overall conclusion is that by both limiting apoptosis and supporting cell cycle progression, mir155 contributes to the continuance of the extra follicular PB response.

In general, the data support the primary conclusion. However, I would consider this a preliminary version of the manuscript that is not yet ready for publication. The conclusions are somewhat overstated. Key data points are not strong enough to draw robust conclusions. To strengthen this manuscript for reevaluation, I would request:

1B, - A more complete phenotypic description of these populations, at least in the wt SwHEL cells, along with a more complete gating hierarchy should be included.

A more complete phenotypic description for the different B-cell subsets has been employed, now showing individual FACS plots. After B220 gating, plasmablast B-cells are defined with the additional marker CD138. Germinal centre B-cells are defined further using the marker FAS/CD95. B-cell blasts are defined within the text.

1D, - Are these from three mice in one experiment, or pooled sera from three experiments? These data are quite important for the studies' focus and I would suggest should be analysed independently at least twice.

The data show sera from independent mice for both miR-155 sufficient and miR-155 deficient SW_{HEL} mice at day 4.5.

2A, - The data are supported, but are shown from a single mouse. Some indication of variation in multiple mice would help support the concluded lack-of-difference in the day 1.5 timepoint vs the established change on days 2.5 onwards.

We have now included quantitation and statistics of the variation in multiple mice at days 2.5 and 3.5 which is now presented as Figure 2B. We hope this is to the satisfaction of the reviewer.

3D - The key conclusion discussed in the results is based on what appears to be a statistically non-significant difference. Also, are these total SwHEL B cells or PB? The results discusses PB but the legend references B cells. Exact #s of PB is relevant in this case. This is important as IRF4, a key PC fate-programming gene is affected by PU.1 which the authors have shown previously to be affected by mir155. A rescue of B cells could feed on to enhance PB #s, but this would not necessarily be PB-intrinsic, even with the caspase and Edu/DAPI profiles shown.

Figure 3D (now 3E) represents the numbers of plasmablast B-cells per 10^6 lymphocytes and this has now been made clearer by addition of further clarification in the figure legend and on the axis of the graph.

In addition to strengthening these current data and tightening the manuscript I would request two additional experiments or justifications. The argument that there is an intrinsic role for mir155 in the PB response, while supported by the data, do not indicate it is PB-intrinsic but rather B cell-intrinsic. An effect on the ability of B cells to activate the perifollicular T cells that support the preGC response (and give rise to PB) would equally feed into a numeric PB phenotype defect. To show a direct intrinsic role, I would suggest that placing the mir155^{-/-} SwHEL cells into competition with ^{+/+} SwHEL cells by co-transfer to eliminate feedback elements that could arise if at the B cell-level, mir155 is impacting the activation of interacting T cells prior to PB differentiation. In this co-transfer context the feedback defects will not compound, such that T cells are equally primed, but defects more directly intrinsic and dependent on the ability of the ^{-/-} B cells to receive T cell help. This could be evaluated by co-transferring CD45.1/.2 and CD45.2/.2 ^{+/+} and ^{-/-} cells into CD45.1/.1 mice that I think could be generated in a 1 or 2 generation cross depending on the SwHEL breeding setup. For the early time points (d1.5/2.5), using two different proliferation dyes (CFSE vs CTV) could reach the same point without any necessary intercrosses.

The defects observed in the absence of miR-155 are not necessarily plasmablast intrinsic as we observe defects in the number of B-cell blasts and also germinal centre cells throughout the course of B-cell differentiation, as shown in Figure 1. Furthermore, the proliferative defect is observed in the earlier B-cell blast population prior to differentiation into plasmablasts, as shown in Figure 2. We cannot comment on whether miR-155 regulates proliferation and apoptosis to the same extent in additional B-cell populations in the SW_{HEL} model, as plasmablast B-cells were the focus of this manuscript.

A second argument is made that it is only in the window at which T cell help is beginning that proliferative differences begin. In addition to more data points for figure 2A, an experiment that would further support this finding would be to demonstrate (or cite) that anti-IgM drives similar proliferative responses in ^{+/+} and ^{-/-} B cells in this window, or test whether non-SwHEL miR155^{-/-} and ^{+/+} NP-Ficoll responses or LPS PB responses are equivalent or not.

We have added more data points for Figure 2A, including the percentage of dividing cells in each generation for days 2.5 and 3.5, with statistical analysis included. In support of this data, a previous publication by our group has shown a slight defect in CFSE dilution in miR-155 deficient splenic B-cells cultured with LPS and IL-4 (Lu et al., 2014) compared to miR-155 sufficient B-cells. In this same experiment, the frequency of CFSE+ CD138+ B-cells was reduced compared to wild type mice. This has been added to the discussion as per the reviewers request.

-- Additional points for clarification:

The text needs to be tightened. There are numerous instances throughout in which the results section are disconnected from the figure legends (for example, the results mentions a green group in Figure 1B, but these cells are grey in the figure and associated legend).

This figure has now been replaced with a more comprehensive-flow cytometry gating strategy

The methods section lacks some details, such as how EdU was administered to the mice.

We have now added information into the methods section regarding how EdU was administered into mice and subsequent analysis.

The gating strategies used, while previously published by other groups, are a little too ambiguous to be fairly evaluated in this study. Specifically, GC B cells are typically identified by FAS, GL7, PNA, Ephrin B1 or BCL6 expression, along with downmodulation of CD38 and IgD; other strategies are possible and often appropriate, such as the putative strategy employed here, being down modulation of BCR relative to activated non-GC B cells, but as far as I can ascertain, the low HEL expression cited to identify GC B cells is actually referencing cytoplasmic HEL staining differences in PB and GC B cells. It is a little confusing to use surface BCR alone in CD138negative cells to distinguish preGC and GC B cells.

We have also used FAS/CD95 to define germinal centre cells and a more detailed gating strategy is now incorporated into the manuscript.

Also, GC cells are essentially irrelevant for this manuscript, whereas clear demarcation of the PB is absolutely crucial. That is, stronger evidence that the distinction between B-blasts and committed GC cells is necessary, if these cells are to be discussed in the results section as in the current version. A phenotypic profile of the analysed populations would help.

We do not understand the above comment and would not be sure how to satisfy the reviewer. However, if the reviewer is questioning the relevance of the germinal centre reaction in this manuscript, this is not the primary focus, as the role of miR-155 in the germinal centre reaction has previously been studied in detail in the SW_{HEL} system in an earlier publication (Nakagawa et al.,

2016).

With the exception of the seemingly appropriate GO expt execution and analysis (Figure 4). many of the experiments that seem well executed lack statistical rigour (Figure 1-3) which makes it difficult to have confidence in the strength of the associated conclusions. There are places in the results section where I view the conclusions as overstated based on the presented data. That is, where strong conclusions are made, the analysed data are not significant at that point (esp. 3C&3D); the reasons such inferences are suitable given the outcome needs to be more clearly justified, or the presented data reconsidered to address the major conclusion that is intended from the experiment. In numerous cases, multiple time points are analysed. The authors have, appropriately, selected non-parametric analyses, yet employ (inappropriately, and without suitable justification) multiple T-tests on these data. *Post-hoc analyses following non-parametric anova are more typical analyses for the field.* **Also, some data are presented on log-axes, yet averages are shown as arithmetic mean + SEM, which provides little information about the distribution of a log-normal population. More appropriate is either geometric mean + geo SD factor or median + IQR. Where non-standard analyses are employed, I request that the authors justify these uses more explicitly. In some such cases, parametric analyses after log-transformation can also be appropriate in place of non-parametric tests. For example, for Figure 1C, the difference is reasonably clear and the expt seems sufficiently powered to reveal the argued difference(s), but a T-test is an atypical and I suggest, an inappropriate test to employ in such a case.**

We thank the reviewer for detailed comments of the statistical tests used throughout the manuscript and the presentation of data. We have revised the analysis of data throughout Figures 1-3 and sought additional input from a statistician. In Figure 1, for plasmablast and germinal centre cell number data, we log transformed the data and then performed a Two Way ANOVA. Plots of the residuals from the model indicated acceptable normality and homogeneity of variances. The interaction term of the ANOVA showed a significant effect of genotype that was dependent on the time. As we were primarily interested a priori in the effect of genotype we then did three orthogonal comparisons of the effect of genotype at each of the time points. Each of these was significant. We did not adjust for multiple testing because these were orthogonal planned comparisons as discussed in Biometry by Sokal and Rohlf. For Figure 1D a Two Way ANOVA with Sidak's multiple comparison test has now been employed, as suggested by the reviewer. For Figure 2 in data sets where the data was not normally distributed, and where we are interested in the comparison of two groups only, a non parametric Mann-Whitney test has been used. For Figures 3E and 3F, we performed a non-parametric Kruksal Wallis with Dunn's multiple comparison test as suggested by the reviewer.

In addition we have added detailed information about statistical analysis to the methods section as follows:

"Statistical analyses were performed in GraphPad Prism software or R Studio, tests are indicated in the figure legends. All data were tested for normality of residuals. If data were normally distributed, parametric tests were employed. For non-normally distributed data, several transformations were attempted and if this was satisfactory, parametric tests were used. Where transformation of data yielded non-normally distributed residuals, non-parametric tests were used. For B-cell blasts number, data were square root transformed. For testing the effects of genotypes on plasmablast and

germinal centre cell number, the data were log transformed and tested by Two Way ANOVA. There was a significant interaction between genotype and time. For CFSE data values were arcsine transformed. For EdU incorporation data, values were arcsine transformed and tested by Two Way ANOVA."

We have considered the presentation of data, as suggested by the reviewer, and where individual data points are presented on scatter plots we have removed the mean and standard error bars as the spread of data is clear.

The claim is made from Fig 2 that by having more cells in G1 and fewer in S/G2/M (Figure 2C), mir155 is impacting progression from G1 to S. However, DAPI staining does not distinguish G1 from G0 cells, and every aspect of the study could be ostensibly explained by a larger fraction of cells in each division becoming quiescent. The data could equally indicate the cells have a reduced division destiny or receive poorer T cell help and are thus prone to quiescence. Also the HEL-SRBC activation system essentially activates the BCR concomitant with transfer, which could induce proliferation before receipt of T cell help. To claim that the loss of proliferation at day 2.5 is related to T cell help, these data would be strengthened by showing more than one representative mouse for each timepoint (Figure 2A), or some in vitro anti-BCR studies in mir155^{-/-} and ^{+/+} B cells, or reference to such studies.

As requested by the reviewer we have added additional time points and analysis to Figure 2A to show more than one mouse. We now show a representative FACS plot in Figure 2A which is complimented by full analysis of the proportion of cells in each division at both 2.5- and 3.5-days post immunisation, now shown as Figure 2B.

The data in 3C & 3D are used to argue that PB are more numerous and frequent in mir155^{-/-}-BCL2-rescued cells. 3C actually shows a proportional reduction in PB in mir155^{-/-}-BCL2 vs mir155^{-/-}, while 3D shows a statistically non-significant increase in (from the legend) B cell numbers. It is further unclear from the text+figure+legend whether these encompass B cells + PB, just B cells, or just PB. While these can be used to back-calculate the # of PB, this is a rather circuitous way to get to the argument that (I think) the authors are trying to make. The key parameter (#of PB) needs to be evaluated directly, the power of the experiment increased, or the experiment redesigned to address the question as discussed and interpreted in the results section. As this is a key argument for the paper, I think the conclusion is overstated.

We thank the reviewer for highlighting this discrepancy. We have now made clear that data in Figure 3D (now Figure 3E) shows the number of plasmablast SW_{HEL} B-cells which have been evaluated directly, as suggested by the reviewer, and does not represent total B-cells. The axis on the graph and also the text in the figure legend has been modified. The representative FACS plot which illustrates the gating strategy for this calculation (now Figure 3D) has been modified with arrows

now present between the CD45.2 donor population and the plasmablast population, all of which are used when calculating the number of plasmablast B-cells.

Minor points:

- In the results/discussion, please cover more explicitly and appropriately reference that the role for miR155 in PB response has been shown rather than is established here, but these studies are confirming the phenotype in SwHEL system.

This is the first time that a role for miR-155 in the survival and proliferation from the stage of B-cell blasts in vivo has been shown. We also state in the introduction and discussion the previous work which has reported a role for miR-155 in plasmablast differentiation.

In the discussion of Figure 1B the authors refer to plasma blast B-cells, and then to extra follicular PB in 1C. I would suggest they reconsider their definition of a PB to the more conventional proliferating cell producing antibody, whereas a B-blast is a proliferating focus of B cells. Making these terms more explicit would help clarify the text.

Plasmablasts in the SW_{HEL} system have been well characterised in a previous publication from Robert Brink's lab (Chen et al., 2009), in addition to expressing CD138 as we show, they are also positive for Blimp-1 and can be found in the extrafollicular space. As such it seems sensible to keep nomenclature consistent between manuscripts. However, we have increased the clarity of the text to ensure the term 'plasmablast' is used rather than 'extrafollicular plasmablast' when describing data.

2A - It would be nice to see what % of the HEL B cells in each case had divided at all (I can't see it in the histogram)

The percentage of dividing cells in each generation has now been quantified for days 2.5 and 3.5, with statistical analysis included, this is now shown as Figure 2B. Furthermore, the lines on the CFSE plots in Figure 2A have been made thinner and clearer.

2B please show representative flow plots of Edu staining on the PB

Representative flow cytometry plots for EdU staining in both miR-155 sufficient and deficient SW_{HEL} plasmablast B-cells has now been added to Figure 2 and additional information is complete in the respective figure legend.

2C please show representative flow plots of DAPI staining on the PB. Stats: These data are paired (ie., repeated measures).

Representative flow cytometry plots for cell cycle analysis with DAPI staining in both miR-155 sufficient and deficient SW_{HEL} plasmablast B-cells has now been added to Figure 2 and additional information is complete in the respective figure legend.

2A-C please state # of expts

The number of experiments carried out for the data in Figure 2 is now added to the figure legend.

The manner in which EdU was administered is not stated, and if it was administered i.p., the time after administration that euthanasia was performed should be stated.

We have now added the method of EdU administration and analysis to the methods section of the manuscript.

In the discussion of figure 2C, I suggest the conclusion overstates the finding. The assays used do not distinguish between G0 and G1 phases; to state that G1/S phase transition is blocked is inappropriate. The conclusion that fewer cells are now in S/G2/M is, however, appropriate.

We have made changes to the conclusion and the following text is now present in the text:

"When DNA was quantified with DAPI, at day 3.5 there was a lower frequency of cells in the S-G2-M stage of the cell cycle in SW_{HEL} Mir-155^{-/-} plasmablast B-cells compared to SW_{HEL} Mir-155^{+/+} plasmablasts and an increase in the frequency of cells in the G1 stage, which became statistically significant at day 4.5 (Figures 2C and 2D). "

We have also removed the G1 data from Figure 3C to focus on the most relevant conclusions of the decreased proportion of cells in the S phase, as commented by the reviewer. We hope this makes for a more convincing argument.

3A please show representative flow plots

Representative FACS plots for cleaved caspase staining in SW_{HEL} miR-155 sufficient or deficient plasmablast B-cells is now added to the manuscript as Figure 3B, with additional text in the figure legend.

3C - showing the individual experimental results would be helpful. Otherwise please state what the average and error bars show. Also please consider if the type of error bar shown is appropriate for DDCT data.

Figure 3C (now 3D) showed raw data in the form of a representative flow cytometry plot for analysis of adoptively transferred SW_{HEL} Mir155^{+/+} or Mir155^{-/-} B-cells expressing a human Bcl2 transgene at day 4.5 post immunisation. Average and error bars are not present on the figure.

3D please state what error bars represent. Please do not use SEM

After seeking input from a statistician, it was decided that the plots we are using are able to clearly show the spread of data and that mean and standard errors bars were not required.

3D these data are difficult to interpret. The most meaningful c.f. would be g2 vs g4, but these are not different. But there is a 2-3-fold shift in average (is it mean here?), which is hard to argue is meaningless, but rather underpowered. This seems to be a key comparison for the study, but I would suggest the strength of the conclusion is overstating the strength of these data. As presented, these data show more (but not statistically significantly more) B cells, but relative to each B cell, a smaller fraction of PB (3C). If the argument is about PB, please use data that more overtly demonstrates the point.

The key conclusion from this data is that there is some rescue in the number of plasmablast B-cells in miR-155 deficient, Bcl2 transgenic mice, but the phenotype is not fully restored. We believe this is meaningful as it suggests that other mechanisms other than apoptosis are playing a role in the phenotype. The data were input into R and normality was assessed. The data was not normally distributed and therefore a non-parametric Kruskal-Wallis with Dunn's multiple comparison test was used.

Also, if all data points are shown, it is not representative, but rather all-encompassing, so please be clear with what your data are reflecting.

The written text for Figure 3 has now been changed in the figure legend at the reviewers request.

3E - these differences are to be expected, yet, there seems a limit on absolute possible number of PB in the wt: if fewer are dying but # is similar, what is happening to them? i.e., did you look at bone marrow for early PC seeding?

The average number of plasmablast B-cells does increase with BCL2 compared to miR-155+/, although this is not statistically significant. We show that there is a decrease in the frequency of active caspases in the cells with the Bcl2 transgene in both miR-155 sufficient and deficient plasmablasts, which is indicative of decreased apoptosis. The small change between miR-155 sufficient mice with or without Bcl2 transgene could be explained by only a small proportion of PBs undergoing apoptosis at this time point, especially in comparison to the miR-155-/- PBs and therefore may explain why we observe less increase in cell numbers than in miR-155-/-.

In the conclusions to Figure 3, they discuss that in addition to rescuing PB from apoptosis, other mechanisms come into play. BCL2 also rescues B cells from apoptosis, so how is it distinguished that this rescue relates to anti-apoptotic effects at the PB level rather than simply as a function of having more B cells available for differentiation?

We would expect to see an increase in the survival of all B-cell population numbers as Bcl2 is expressed in all B cells. Indeed we observe an increase in the number of miR-155 sufficient plasmablast B-cells in Bcl2 transgenic mice, albeit this is not significant. However, we observed a much larger increase in miR-155 deficient plasmablast B cells with huBCL2 than we observe with WT cells at day 4.5 post immunisation when the defect is most severe, suggesting that more apoptosis is occurring. In addition, we observe a decrease in the frequency of active caspases specifically in the

plasmablast population in Bcl2 transgenic mice at day 4.5, indicating that there is reduced apoptosis. However, we cannot rule out that an increase in B-cell blast number in Bcl2 tg mice has some impact on the increased number of plasmablast B-cells, as we observed defects in cell number in the absence of miR-155 from day 3.5 onwards, including in the B-cell blast population. However if this was the case, the data show that the plasmablast cell number still could not be fully rescued and therefore still indicates a requirement for miR-155 in the survival of B-cell blasts and plasmablast B-cells.

Reviewer #3 (Comments to the Authors (Required)):

This manuscript from Leyland et al., appears to be an interesting story about the contribution of mir155 to plasmablast survival and proliferation. However, the authors have been quite careless in the process of putting their data together, making assessment a little difficult. I have got the gist of the story nonetheless and will likely be positive if some attention is paid to making the data clearer as pointed out below:

On page 5 the authors state that plasmablasts, defined as being B220^{lo} HEL BCR⁺ and CD138⁺, were represented in green in Figure 1B. In Figure 1B there is no indication of the CD138 gate and the plasmablasts are hard to distinguish with the colors used. They are in "grey" not "green" but this figure needs to be made clearer.

Gating strategies for the different B-cell subsets has been employed, now showing individual FACS plots. After B220 gating, plasmablast B-cells are defined with the additional marker CD138. Germinal centre B-cells are defined further using the marker FAS, as suggested by the reviewer. B-cell blasts are defined within the text.

In Figure 2 we are apparently looking at CFSE dilution in plasmablasts but how these cells were identified is not clear in the text, the legend or most importantly in the figure itself.

We thank the reviewer for highlighting this. In Figure 2, CFSE dilution was assessed in B-cell blasts which precede differentiation into plasmablasts. We have now added how these cells were identified in the figure legend. This now mirrors what is also written in the text as such:

"SW_{HEL} Mir155^{+/+} (black line) or Mir155^{-/-} (grey line) HEL binding, B220⁺ B-cell blasts were assessed for CFSE dilution."

In Figure 3 it appears that CD138 is no longer used to gate on plasmablasts -at least not based on the statement in the legend.

We show in Figure 1 that HEL⁺ B220^{lo} plasmablast B-cells are also CD138⁺. These cells have also been well characterised by Robert Brink's lab in a previous publication (Chan et al., 2009) where it is also shown that HEL BCR⁺ B220^{lo} plasmablasts are positive for Blimp-1 expression and can be visualised in the extrafollicular space in the spleen.

In figure 3C the CD45.2 gate is clear. No arrows to indicate that the right side panels are derived from the CD45.2 box. Assuming that this is so, the B220 low box on the right presumably represents plasmablasts but there is no indication provided in the legend or the figure as to what is in each box in this figure. What cells occupy the oval demarcations on the right of this box in Fig 3C is also not made clear.

We have made the FACS plot in Figure 3 clearer by adding arrows between the CD45.2 donor population and the plasmablast population. Additional text has also been added to the figure legend to further clarify the gated populations in the plots as such:

"Representative FACS plot showing the gating strategy for analysis of adoptively transferred SW_{HEL} Mir155^{+/+} or Mir155^{-/-} B-cells expressing a human Bcl2 transgene at day 4.5 post immunisation. Plots were previously gated on lymphocytes and single cells and analysed for CD45.2 donor cells and HEL⁺ B220^{lo} plasmablast B-cells. HEL⁺ B220^{hi} germinal centre B-cells could also be visualised in all mouse strains."

April 12, 2019

RE: Life Science Alliance Manuscript #LSA-2018-00244-TR

Dr. Rebecca Leyland
Babraham Institute
Babraham Research Campus
Cambridge CB22 3AT
United Kingdom

Dear Dr. Leyland,

Thank you for submitting your revised manuscript entitled "MicroRNA-155 is essential for the optimal proliferation and survival of plasmablast B cells". As you will see, the reviewers appreciate the introduced changes and we would thus be happy to publish your paper in Life Science Alliance pending final revisions necessary to meet our formatting guidelines:

- please upload all figures as separate files
- please fill in the electronic license to publish form

A. FINAL FILES:

B. MANUSCRIPT ORGANIZATION AND FORMATTING:

Sincerely,

Reviewer #1 (Comments to the Authors (Required)):

In "microRNA-155 is essential for the optimal proliferation and survival of plasmablast B cells" by Leyland et al. explore the role of miR-155 in the humoral immune response, expanding previous studies by the same authors. In its original form, the manuscript was missing some clarifications,

including statistical analysis. The authors have now fulfilled many of the reviewers' requests, which has very significantly improved the robustness of the data and the clarity of the text and figures. Thus, the claims are now well supported by the data presented.

Reviewer #2 (Comments to the Authors (Required)):

The manuscript is substantially improved. The major conclusions are supported by the findings and the work is now suitable for publication in LSA.

April 30, 2019

RE: Life Science Alliance Manuscript #LSA-2018-00244-TRR

Dr. Rebecca Leyland
Sheffield Hallam University
Biosciences and Chemistry
BMRC
Howard Street
Sheffield S11WB
United Kingdom

Dear Dr. Leyland,

Thank you for submitting your Research Article entitled "MicroRNA-155 is essential for the optimal proliferation and survival of plasmablast B cells". It is a pleasure to let you know that your manuscript is now accepted for publication in Life Science Alliance. Congratulations on this interesting work.

DISTRIBUTION OF MATERIALS:

Again, congratulations on a very nice paper. I hope you found the review process to be constructive and are pleased with how the manuscript was handled editorially. We look forward to future exciting

submissions from your lab.

Sincerely,
